# Management and Feeding Strategies in Early Life to Increase Piglet Performance and Welfare around Weaning: A Review

**DOI:** 10.3390/ani11020302

**Published:** 2021-01-25

**Authors:** Laia Blavi, David Solà-Oriol, Pol Llonch, Sergi López-Vergé, Susana María Martín-Orúe, José Francisco Pérez

**Affiliations:** Department of Animal and Food Sciences, Animal Nutrition and Welfare Service, Universitat Autònoma de Barcelona, 08193 Bellaterra, Spain; David.Sola@uab.cat (D.S.-O.); Pol.Llonch@uab.cat (P.L.); Sergio.Lopez.Verge@uab.cat (S.L.-V.); Susana.Martin@uab.cat (S.M.M.-O.); JoseFrancisco.Perez@uab.cat (J.F.P.)

**Keywords:** feeding, management, strategies, weaning, pigs

## Abstract

**Simple Summary:**

Weaning is an important period for the swine industry and is influenced by the early events that occur during gestation and lactation. Therefore, a range of dietary and management strategies have to be implemented to achieve optimal health status, maturity, and weight at weaning. In this review, we aimed to identify the major dietary nutrients and management strategies to enhance fetal growth, reducing the oxidative and inflammatory status of sows, modulating the microbiota of sows, enhancing colostrum and milk production, and taking care of neonatal piglets.

**Abstract:**

The performance of piglets in nurseries may vary depending on body weight, age at weaning, management, and pathogenic load in the pig facilities. The early events in a pig’s life are very important and may have long lasting consequences, since growth lag involves a significant cost to the system due to reduced market weights and increased barn occupancy. The present review evidences that there are several strategies that can be used to improve the performance and welfare of pigs at weaning. A complex set of early management and dietary strategies have been explored in sows and suckling piglets for achieving optimum and efficient growth of piglets after weaning. The management strategies studied to improve development and animal welfare include: (1) improving sow housing during gestation, (2) reducing pain during farrowing, (3) facilitating an early and sufficient colostrum intake, (4) promoting an early social interaction between litters, and (5) providing complementary feed during lactation. Dietary strategies for sows and suckling piglets aim to: (1) enhance fetal growth (arginine, folate, betaine, vitamin B_12_, carnitine, chromium, and zinc), (2) increase colostrum and milk production (DL-methionine, DL-2-hydroxy-4-methylthiobutanoic acid, arginine, L-carnitine, tryptophan, valine, vitamin E, and phytogenic actives), (3) modulate sows’ oxidative and inflammation status (polyunsaturated fatty acids, vitamin E, selenium, phytogenic actives, and spray dried plasma), (4) allow early microbial colonization (probiotics), or (5) supply conditionally essential nutrients (nucleotides, glutamate, glutamine, threonine, and tryptophan).

## 1. Introduction

The average litter size of sows has been increased by genetic selection over the last decades; however, this is associated with a reduction in the average birth weight and, concomitantly, an increased within-litter body weight (BW) variation. Consequently, this has increased the proportion of small piglets (less than one kg birth weight) in large litters [1]. Some of the pigs with low birth weight (LBW) exhibit consistent long-term effects, such as higher morbidity–mortality and lower growth rates [2], which could suggest that there are long-term alterations due to the negative effects of dietary restriction during pregnancy, among other causes. This growth failure is explained partly by the reduced intestinal size, shorter villus height and the villus:crypt ratio [3], and the lower intestinal trophic responses to the introduction of enteral feed during the early postnatal period. Changes in gut microbiota and signs of an immature innate immunity, are also evident in pigs with LBW [4].

In addition, weaning weight is also important in post-weaning growth, morbidity, and mortality. Lighter piglets (less than 5.0 kg) are a significant cost to the system due to reduced market weights and increased barn occupancy; however, they also represent the greatest marginal opportunity to decrease days to slaughter (8 days more to achieve 125 kg of BW or weigh 3.3 kg less than piglets weaned at 5.5 kg BW [5]). Stein [6] also reported a significant performance and mortality penalty for groups with very light (<4.5 kg) average weaning weight and a substantial penalty, even for groups with light (5 kg) average weaning weights. He also observed that there was a positive effect of weaning weight on average daily gain, i.e., the higher the initial BW, the higher the average daily gain.

Several studies conducted in our group showed some insights into the effect of birth BW and weaning BW until slaughter (see Figure 1). In initial growth phases (i.e., until the end of lactation), changes in pigs’ BW category are common, reflected by low correlation coefficients, which means that a certain percentage of piglets change category over the period. From weaning onwards these changes are less common (correlation coefficients are higher), meaning that pigs that reach nursery with a low BW category have a higher probability of remining in the same class at the end of the cycle, which negatively affects the margin of growth improvement achieved by implementing different strategies.

Recently, Montoro et al. [8] showed similar findings, as they found that birth BW has a low capacity to act as a predictor of subsequent growth until slaughter, mainly because BW categories can change throughout lactation. Furthermore, similar results were observed when suitable and practical indicators of early carcass depreciation were determined, and the conclusion was that BW could be a valid predictor at the end of nursery [7].

Therefore, it is very important to reach a high BW at weaning or even at the end of nursery, because from this point onwards, the opportunity to implement strategies so that lighter pigs can successfully catch up to their heavier pen mates decreases dramatically. The extreme case would even be beyond the nursery (i.e., growing-finishing period) when the correlation is particularly high (ρ = −0.80/−0.90). This means that the BW at this point will remain in the same category until slaughter [2].

Consequently, the present review explores the main factors affecting both birth BW and weaning BW. These factors include intrauterine growth, colostrum intake, microbial early colonization, and milk yield during lactation. We describe early strategies (management and feeding strategies), either focusing on the sows or the piglets during lactation, which aim to decrease the proportion of small piglets at weaning.

## 2. Factors Affecting the Weaning BW

### 2.1. Birth BW

High producing sows (>30 piglets weaned/sow/year) deliver large litters with an increased percentage of pigs with LBW. Up to 30% of a litter have been exposed to various degrees of intra-uterine growth restriction (IUGR) [9,10].

Moreover, higher mortality during the first days of life is reported for both LBW and IUGR pigs [10]. IUGR is usually defined as impaired growth and development of the fetus and/or its organs during gestation [11], and because of the brain-sparing effect [12], IUGR pigs can even be recognized by their head shape [9,10,13]. However, LBW pigs are considered pigs with less than 1.25 or 1.0 kg at birth [14,15,16], but without organ growth restriction. Independently of the BW at birth, those animals classified as IUGR pigs show poor seeking and sucking behavior after birth (defined here as physical strength or vigor) and may not show enough vitality to ensure proper colostrum intake, with the corresponding negative impact on the subsequent health status and mortality within the first five days after birth [17]. It is recommended that pigs take at least 250 g of colostrum within the first 24 h to survive [18], but IUGR piglets often consume only 100 g of colostrum per kg of BW or less within the first 24 h [9]. Moreover, the low glycogen stores reported for IUGR piglets may also explain their low newborn vitality [19,20]. All these effects directly affect the concept of “fetal programming” that may be a determinant for neonatal pig survival or even for further pig growth and efficiency [21].

The relationship between maternal nutrient intake during pregnancy and growth of the fetus is extremely important in determining pregnancy success, life-long health, and the productivity of the newborn [22,23]. The size and nutrient transfer capacity of the placenta (considering that there is no blood contact between sow and fetus during gestation [24]) play a central role in determining the prenatal growth of the fetus and are directly related to birth weight and IUGR. Trans-placental exchange is dependent upon uterine and umbilical blood flow, and these, in turn, are largely dependent on the adequate vascularization of the placenta. This is why nutrients with vasodilator capacity have a positive impact on the growth of the fetus.

The growth of the placenta precedes that of the fetus and a strong positive association exists between placental mass and the piglet size at farrowing [25]. Aiming to promote placenta development, it has been reported that increasing the uptake of amino acids in the bloodstream may promote changes in the adhesiveness capacity of the trophoblast cells and form out growths [26]. This effect is mediated by the mTOR signaling [21]. Moreover, the positive regulation of mTOR in fetuses has also been related to secondary fiber development in pigs, exerting the above mentioned “fetal programming” [27]. Proteins related to energy supply, protein metabolism and muscle structure, function, and proliferation are expressed differentially in IUGR-affected piglets [28], which indicates impaired metabolism and reduced muscle growth and development in these piglets. Thus, IUGR pigs have economic consequences for the subsequent production efficiency, such as reducing the gain to the feed ratio and decreasing the percentage of meat [29] and increasing carcass body fat [30].

Fetal growth occurs rapidly from day 77 of pregnancy until term [31]. Therefore, it is important to feed sows correctly during this time period. It has been reported that feeding a very low protein diet (6.1 to 6.5% CP (crude protein)) during late gestation results in a reduction of the piglet’s birth BW [32] and can lead to IUGR [33,34]. This is possibly due to a reduction in indispensable amino acids, which leads to a damaged lipoprotein metabolism [35]. However, the energy levels fed during gestation influence the sow’s performance more than piglet’s BW. Low levels of energy during gestation are related to lower body fat reserves at farrowing or at weaning and more days to return to estrus [36], while high energy intakes increase the sow’s weight and body condition [36], but do not influence the piglet’s birth and weaning BW or survival [37,38].

In relation to the sows’ feed intake level, it has been reported that increasing feed intake (approximately 20%) 20 days before farrowing increases the sow BW gain, but has no effect on piglet and litter quality (individual BW and total litter weight) [39]. This has also been directly related to an impaired appetite and reduced feed intake [39]. Finally, a bumped feed intake before farrowing seems to be associated with an increased energy and amino acids (AA) intake, which promotes sows becoming overweight but does not have a significant impact on litter quality even in high producing sows [39]. This is directly related to the following section because bump feeding may directly affect colostrum and milk production.

### 2.2. Colostrum and Milk Production

Colostrum has an important function as it provides the pig with maternally-derived, passive immunity after birth. It is also the main source of nutrients and growth promoting peptides for intestinal tract development in the newborn, strongly contributing to passive immunity [40,41,42]. Compared to regular milk, sow colostrum contains high concentrations of nutrients (protein and fat) and immunoglobulins (IgG, but also IgA, IgM), immune cells, and various antimicrobial substances such as lactoferrin [43,44]. Therefore, colostrum plays an essential role in pig survival and growth and is the earliest contribution to reducing neonatal diarrhea. While colostrum is freely available during the first hours after birth, in modern sows, access to colostrum might be limited due to the large number of piglets born [45]. Therefore, colostrum production and colostrum intake are one of the most challenging aspects for high production breeds. Unlike milk production, which is largely dependent on the demand by piglets and litter size, colostrum production is limited (estimated to vary between 2.5 and 5.0 kg over 24 h for a litter of 8–12 piglets) and highly variable between sows. The variability in sow colostrum production is not only determined by nutritional aspects, but is also affected by: (1) sow parity number: young sows until 3–4 farrows produce more colostrum than old sows, (2) mammary glands development, and (3) endocrine status: individual variations in progesterone and prolactin response, a delay in the exchange of these hormones before farrowing may dramatically reduce the yield of colostrum. Moreover, piglet vitality may limit their ability to suckle colostrum. As litter size increases, a higher number of low-weight and low-vitality piglets [1,16,46], with a lower colostrum intake, is being reported.

Previous reports suggest that colostrum yield is positively related to the litter size and piglet vitality as well as with milk yield [47]. This indicates that maximizing the colostrum yield can both increase piglet survival and enhance milk production. It is known that piglet suckling stimuli, as they are massaging, stimulating, and draining the teats, enhance local blood flow together with hormonal and nutrient release, thus increasing the milk production of the teat [48]. Therefore, the onset of lactation and transition milk yield (yield from 36 to 60 h after parturition) is positively influenced by the litter size 24 h after parturition [47]. The correlation is maintained throughout the four weeks of lactation, which is probably related to a higher number of milk-producing glands and more efficient milk synthesis stimulation in large litters [49].

However, milk production may also depend on the metabolic status of the sow. It is widely accepted that fetal growth during late pregnancy, and the synthesis of colostrum and milk during lactation lead to the catabolic status of the sows [50], the production of reactive oxygen species (ROS), and the induction of oxidative stress [51]. Thus, sows may experience increased oxidative stress and inflammation during the gestation and early lactation period [52], which could affect the mammary epithelial cells. Wang et al. [53] researched the role played by oxidative stress in sows in high or low litter performance during lactation, backfat thickness, number of piglets, and litter weight. Differences in the sow’s antioxidant level were observed at the initiation of the lactation period, when serum antioxidant levels were higher and thiobarbituric acid reactive substances (TBARS) were lower at d 1 of lactation in the group of high litter performance sows. This suggests that ROS in sows may significantly damage mammary epithelial cell, and consequently affect milk yield performance [54]. A beneficial antioxidant capacity in sows could attenuate oxidative stress-related effects on milk yield, which opens up opportunities to guide nutrient interventions to alleviate oxidative damage.

Piglet vitality and activity within the first five days after farrowing are determinant of the switch-on of milk production and proper piglet growth [17]. It has been reported that, in equal litter sizes, lower litter BW, total litter BW gain, and individual piglet BW were directly related to litter activity [17]. The negative impact of heat stress on litter BW seen at weaning had already occurred at day seven of lactation. As sows had similar daily feed intakes, the litter behavior itself may help to explain these differences. Therefore, litter apathy due to heat stress or other issues impairs a proper switch-on of lactation in summer [55].

### 2.3. Gut Microbiota

It is generally recognized that the process of microbial colonization of the gut after birth plays an important role in the development of the neonatal immune system of mammals with implications during their whole life. In humans, this window of time has been established as 100 days [56], a period during which dysbiosis and shifts in specific bacterial taxa have been related to an increased risk of asthma in future life [57]. While something similar is expected to occur in pigs [57,58], further research is needed to understand the impact of early life events on the piglet’s microbiome and its potential relevance for the pig industry [59,60,61]. Some authors have described how different exposure levels to stress or the use of antibiotics can determine changes in the gut microbial colonization of pigs eight days after birth with implications for immunity development [62]. Some evidence has been published that defines differences in the fecal microbiota of pigs at as early as seven days of life, determining their susceptibility to suffering post-weaning diarrhea four weeks later [63], and emphasizing the potential of the early establishment of the microbiota for the development of the immune response. The relationship between the IUGR syndrome and the microbiota has also been evidenced, showing how microbiota colonization is significantly altered in IUGR pigs during the first 12 h of birth [64]. Li et al. [65] also highlighted the role played by the microbiota in this syndrome demonstrating that LBW pigs have a different fecal microbiota compared to normal-birth-weight pigs during early life with a lower abundance of *Lactobacillus* and other altered bacterial communities.

Undoubtedly, the sow is the main and first donor of fecal microbiota to pigs, playing a relevant role in this early process of microbiota establishment. Recent studies on administering maternal fecal microbiota to neonatal pigs have demonstrated that this early intervention can improve the growth performance of pigs, decrease intestinal permeability and stimulate SIgA (Secretory Immunoglobulin A) secretion, modulating gut microbiota composition [66]. The importance of the mother-effect defining a particular microbiota composition in the nursing piglet was also evidenced by Mu et al. [67], who analyzed the early-life microbiota succession in pigs using a cross-fostering piglet model. Therefore, maternal environmental factors such as diet composition and antibiotic treatment can induce changes in maternal microbiota and may have huge effects on offspring gut physiology [68]. The role of the sow as an early donor of gut bacteria to the pig could even start before birth [69]. Until recently, the environment of the mammalian uterus and also the mother’s milk were thought to be sterile. However, pioneer studies have shown that animals are exposed to microorganisms even before birth [70] and that, in addition, breast milk in humans has been shown to be capable of delivering certain intestinal microorganisms from the mother to the offspring using the entero-mammary route [71]. Thus, the mother has some control over the microorganisms to which the piglet is exposed in order to determine the correct programming of the newborn.

Considering the pivotal importance of the microbiota for young animals, favoring the establishment of a complex and well-balanced intestinal ecosystem in the pig’s intestine is currently recognized as a key point in any antibiotic reduction program [72]. Currently, many of the new non-antimicrobial alternatives, such as zinc oxide, phytogenics, and pre or probiotics are being evaluated in relation to their potential to restore intestinal balance and to improve the adaptation of piglets to critical weaning stress [73]. The use of alternative unmedicalized diets based on including fibrous by-products and functional feed ingredients has also been proposed as a way of decreasing the use of in-feed antimicrobials in the control of post-weaning diarrhea, achieving faster development of a mature and resilient intestinal microbiota [74].

One of the main causes of an impaired weaning weight and increased batch variability during the nursery period is the high prevalence of digestive pathologies associated with opportunistic pathogens. Between them, *Escherichia coli* is the main pathogenic agent (some particular strains) implicated in the neonatal and post-weaning diarrhea [73,75] and is responsible for the high use of antimicrobial during this stage. In recent years, different feeding strategies, and particularly different in-feed additives, have been explored as a means of fighting this pathogen and reducing diarrhea in piglets [76,77,78]. However, although there has undoubtedly been a remarkable advancement in this field, there is still a way to go.

## 3. Management Interventions

### 3.1. Sow and Piglet Management

#### 3.1.1. General Management

The literature does not usually consider the general management implemented in the farm as a main factor explaining differences in pig performance. However, decisions on herd management (i.e., one-week batch or batches every two, three, or four weeks) could have an impact on the farrowing period, increasing the number of days that elapse from the first to the last piglet born. The size of the standard growing-finishing facilities in big multiphase production companies, which drives the whole production flow, can usually accommodate one thousand pigs or more. Weekly management systems are the most used. However, farmers may be forced to apply three or four-week herd-batch management systems to achieve the targeted amount of piglets weaned [79] to fill the growing-finishing barns when the sow herd is relatively low (less than 500 sows). López-Vergé et al. [80] carried out an observational study to evaluate the effect of lactation length on the pig’s later performance in a commercial farm operating with a four-week herd-batch system. It was observed that there was a range of up to 10 days of lactation length between piglets of the same batch, leading to large differences in weaning weights in favor of the piglet’s with more accumulated days in lactation. The relevant aspect is that those differences in BW were significantly maintained until slaughter due to better growth performance. Mortality during lactation was also reduced as the lactation length increased, in accordance with the previous findings of Alexopoulos et al. [81].

In European standard intensive conditions, weaning (on average at 28 days) is performed on a single day [82], and leads to differences in weaning age depending on the farrowing day. This difference in lactation length could be even more problematic in hyper-prolific sows due to a lower average birth weight of their offspring. Therefore, it is necessary to reinforce the caring strategies for the lighter piglets of the batch (delay the weaning age). Therefore, as milk constitutes the most beneficial and balanced source of nutrients for piglets [83], differences in lactation length (due to management) can lead to a reduced early performance of the piglets [16] that could last until slaughter [84].

#### 3.1.2. Management during Gestation

In the European Union, since January 2013, all sows have to be housed in groups from week four after mating (confirmed gestation) until one week before expected farrowing (Directive 2008/120/EC) (https://eur-lex.europa.eu/legal-content/EN/ALL/?uri=CELEX%3A32008L0120). The directive also regulates that sows should have access to rooting materials, and outlines floor and space requirements. The new housing system for sows in groups is an improvement for animal welfare, and recently, it has been published that group housing has no negative effect on sow productive performance, reproductive performance, and colostrum composition compared to sows in individual stalls [85]. In addition, there is no evidence that sows in groups have more disease problems than in individual housing [86].

However, having sows in groups may lead to several problems: (1) it has been reported that it can affect fertility due to chronic stress and low feed intake [87]; (2) higher prevalence of lameness, which is related to floor space per sow, the group size, and pen design and flooring [87,88]; (3) aggression between sows, due to competition for access to a limited resource or to establish the hierarchy [87,88]; and (4) it is hard to uniformly control sow body condition and sow weight gain due to dominant sows consuming more than subordinate sows [89]. Moreover, sows that have a low back-fat gain at the beginning of gestation (first third period) are at risk of having a pregnancy failure due to feed competition [90]. The previous problems cited (feed intake control, locomotor problems, aggressions and individual variability) can be alleviated by manipulating management, feeding, and environment (floor, bedding, and design). Familiar sows engage in less aggression [91]; therefore, mixing sows that have been housed together in previous gestations may reduce aggression. Other aspects to implement in order to avoid problems could be: gradual familiarization of unacquainted animals; providing sufficient space and pen structure during initial mixing minimizing opportunities for dominant sows to steal food from subordinates; spatial separation between sows with visual or physical barriers and stalls; and providing a good quality floor, environmental enrichment, and use of straw bedding [87,88,91].

In addition, feeding systems can affect aggression and stress due to competition for feed [87,91]. There are two types: simultaneous and sequential (electronic sow feeders (EFS), Fitmix, Mannebeck Landtechnik GmbH, Schüttorf, Germany.) feeding systems. In a simultaneous feeding system, subordinates sows need protection, which can be achieved by individual feeding stalls or troughs with decreasing lengths of barriers to separate feeding places [92], while in sequential feeding systems protection is necessary while animals are waiting to access the feeder [87]. This can be achieved by training sows how to use the EFS [93]. In addition, a big difference between feeding systems is whether the sow can be controlled individually, such as when identification systems and EFS are used. In a recent study by Jang et al. [94], they observed that sows housed in groups with EFS showed a higher growth performance and survival rate of piglets and tended to have higher BW and back fat thickness than sows kept in stalls. However, sows in ESF had a higher incidence of scratch and locomotion disorders, due to persistent fighting around ESF machines and inadequate bedding materials. Therefore, management during gestation appears to be important as taking care of newborn piglets in order to achieve successful weaning.

#### 3.1.3. Management at Farrowing

##### Farrowing Assistance and Synchronization

Gestation length varies among sows and the induction of parturition makes it possible to synchronize farrowing [95]. In addition, farrowing is difficult to predict and it often occurs at night when staffs are not present. In that case, synchronization may increase the probability that farrowing occurs during day hours, when staff are more likely to be present and provide assistance. Induction can improve farrowing supervision and it also makes early fostering easier [96]. In addition, agents used for induction (i.e., prostaglandin) are known to stimulate colostrum production [97].

The induction of farrowing is usually carried out by administering natural prostaglandin (PGF2α), or a synthetic analogue such as cloprostenol, prior to the expected date of farrowing. Oxytocin may also be given to start the farrowing process. Oxytocin is typically administered between 20–24 h after prostaglandin injection to stimulate uterine contraction. However, the use of oxytocin to trigger parturition has been shown to be erratic and different studies suggest contradictory results in terms of the dosage and side-effects (reviewed by Kirkden et al. [96]). For instance, some authors have observed that its effectiveness is positively correlated with dosage, whereas others have found the opposite effects [98,99].

Induction of parturition has benefits but also risks for the welfare and productivity of the pig farm. For example, if prostaglandins are administered too early before farrowing it increases the risk of premature piglets, which may reduce their viability. Another risk associated to induction is dystocia. This is specifically associated with the administration of oxytocin following prostaglandin treatment [100], with evident harmful effects on the welfare of the sow and the viability of the offspring.

##### Pain Management during Farrowing

Farrowing is an intrinsically risky process for both the mother and the young. Injury, trauma and inflammation associated with parturition (particularly in dystocia parturitions) can have large negative effects on health and welfare, which may ultimately affect productivity. In sows, farrowing has been divided into three separate stages, each of which can lead to pain for the dam. The first stage, mostly regulated by estrogens, includes dilation of the cervix, myometrial contractions and the fetus starts the placement for expulsion [101]. During this phase, visceral pain predominates, stimulated by the mechanical distention of the uterus and cervix dilation. Subsequently the second stage starts with abdominal contractions and expulsion of the fetus, which leads to somatic pain due to distention of pelvic structures such as the pelvic floor and perineum. The third stage includes the expulsion of the fetal membranes and the end of myometrial contractions.

In swine, there are several factors that may modify the degree of pain caused by parturition, such as parturition difficulties and parity, which are the most notorious ones. Parturition difficulty is commonly referred to as dystocia, which results from prolonged parturition or severe assisted extraction and is associated with high levels of pain [102]. Farrowing duration is a critical aspect of dystocia that triggers pain. In normal situations, farrowing duration (from first to last piglet) may average 2.5 h, and parturitions longer than 3 or 4 h [103,104] are thought to be more painful. At the same time, duration of farrowing may be influenced by intrinsic factors of the sow such as breed, litter size [105], higher birth weight [106] and parity; and extrinsic factors, such as the ambient temperature in the farrowing crates [107]. Primiparous females have longer durations of parturitions and the effort associated with it is usually greater than in multiparous females [101] leading to greater pain. Therefore, it makes dystocia more frequent in primiparous compared to multiparous dams [108]. Moreover, a high ambient temperature (25 °C vs. 20 °C) around farrowing may make sows unable to perform thermoregulatory behavior [107]. Sows exposed to the high temperatures around farrowing spent a higher proportion of time lying in the lateral position and had longer farrowing times. They also showed higher rectal and udder surface temperature from the day of farrowing to day 3 after farrowing, which had a negative impact on feed intake and piglets weaning weight. The effects of farrowing duration go beyond the sow, and recent evidence suggests that a longer farrowing is associated with a decrease in sow colostrum yield [109]. This suggests that the negative effects of dystocia (such as pain and prolonged parturition) may also affect the offspring immune competence. It also confirms the findings of Mainau et al. [110] who stated that the administration of an analgesic (i.e., meloxicam) to prevent pain during farrowing increased the concentration of IgG in the serum of piglets and enhanced their pre-weaning growth.

In addition to pain, stress can also be triggered during parturition, affecting the sow’s welfare, but also impacting the productive capacity of the offspring. Farrowing is associated with increased plasma cortisol concentrations [111]. Increases in plasma cortisol could be a response to the intrinsic stress of farrowing, such as pain or novelty (e.g., neonate movements) [112]. In addition, as in intensive production, sows are kept in farrowing crates, physiological stress is exacerbated due to physical restriction of natural behaviors, such as nest building behavior or the inability to avoid piglet requests for nursing [113]. Stress can disrupt farrowing through an opioid mediated inhibition of oxytocin secretion [114]. As evidence of the negative effect of farrowing crates, Oliviero et al. [113] showed that crated sows had lower oxytocin values than sows in pens with some degree of movement. As oxytocin plays a key role in sustaining an optimal level of myometrial contractility, lower levels of oxytocin might result in the prolonged delivery of piglets [113]. In addition to this, extended duration of farrowing and post-partum pyrexia increases the risk of mastitis-metritis-agalactya syndrome [115].

Prolonged or difficult deliveries (without including caesarean section) are associated with increased offspring mortality in sows [116]. The percentage of stillborn piglets ranges from 3 to 12% [102] and accounts for 30 to 40% of the total neonatal mortality [117]. Stillbirth is commonly defined as a newborn that dies just prior to, during, or within 12 h of parturition. The causes of death in stillbirth piglets may be severe acidosis because of oxygen deprivation during an assisted delivery, with subsequent effects on the function of vital organs and overall vitality. Reduced vigor, poor thermoregulation, failure of passive transfer of immunity, poor performance and greater susceptibility to infections are also important secondary problems associated with neonatal asphyxia and acidosis [116].

The administration of non-steroidal anti-inflammatory drugs after parturition should reduce the associated inflammation and pain, and improve the dam’s health and welfare. In sows, Haussmann et al. [118] found a reduced number of body position changes 48 h after farrowing in sows treated with butorphanol and hypothesized that this fact may lead to a decrease in crushing rates. In sows with mamitis-metritis-agalactya syndrome, meloxicam together with an adequate antibiotic treatment was found to reduce the mortality rate in piglets of dystocic parturitions [119].

In summary, pain during farrowing negatively affects the welfare of sows, but it can also reduce the ability to adapt to the environment and their growing capacity. Therefore, prevention of pain should be considered when seeking an improvement in the productivity and welfare of farrowing sows.

#### 3.1.4. Caring for Piglets during the First Days

Swine husbandry generally recognizes that piglets require a warm, dry environment for survival, especially during the first few days after birth. For pigs, birth marks a sudden decrease in body temperature as the starter signal to boost their own thermogenesis. However, pigs are born with very small energy depots. Without brown fat, the glycogen depots are able to supply energy for approximately 16 h [19], and energy supplied from colostrum must contribute to ensure piglet survival because transient milk is not secreted until approximately 34 h postpartum. In these conditions, newborn piglets are in a negative energy balance immediately after birth because of their high physical activity and high-energy needs for thermoregulation [120]. Piglets rely on colostrum, and if there is not enough colostrum, the piglets may die due to hunger or weakness. This is even worse in lighter piglets of the litter, which are usually less vital, which negatively affects their survival because of their higher thermoregulatory issues, and also because they have more difficulties in approaching the udder and suckle colostrum properly [46,121]. Moreover, less vital piglets are at a higher risk of dying by crushed by the sow when they approach the udder [117,122], compared to the rest of the piglets in the litter. Newborn pigs from the same litter compete for the anterior and the middle mammary glands. This could be because the caudal mammary glands produce less proteins than cranial ones [123]. Consequently, supplementation of colostrum to piglets is an extended practice in pig production, particularly to support low BW piglets. There are several management strategies that can be applied to ensure proper colostrum intake, established at a minimum of 200 g per piglet [124]. It is well reported in the literature that manually positioning piglets close to the udder shortly after birth reduces mortality [125,126,127]. Muns et al. [128] highlighted the benefits of orally supplementing the weaker piglets with colostrum recently milked from sows from the same herd, which were higher in IgG blood levels, leading to a better body temperature recovery of the piglet, and better litter performance in the first 24 h. The practice of “split suckling or split nursing” (withdrawing the larger piglets or those that have already taken colostrum for a short period of time) [89,129] just before cross-fostering has also been shown to be effective in guaranteeing colostrum intake of less vital piglets. This practice can result in an improvement in the daily gain and homogeneity of the litter at weaning [130]. Finally, the administration of colostrum replacers available in the market is another strategy usually implemented in farms [131].

After colostrum intake has been guaranteed, the practice of cross-fostering is another common strategy, especially in large litters [132]. Its purpose is to match the different litters in the number of piglets while reducing the variability of each litter at birth [133,134] as well as the pre-weaning mortality [128,134]. The birth weight variability has been found to be related to pre-weaning mortality [99]; however, according to some authors [15,16,18], the specific cause is not the variability itself, but the higher number of piglets with the low BW that this entails. One important issue regarding the practice of cross-fostering has to do with the number of protocols that can be applied; from just matching the sows by the number of piglets to a very complex scenario in which sows are classified by productive potential and piglets are distributed, by number and size, among the available sows. In practice, intermediate protocols are usually chosen. As cross-fostering is a topic widely reviewed in the literature (see Muns et al. [132,135]) the present review will only focus on some recommendations that can be considered: (1) first, it is important to identify viable and non-viable piglets and have the expertise to assign the smallest piglets to the best sows [128]. This is especially important in highly prolific sows, when the number of piglets is usually higher than the number of functional teats; (2) cross-fostering should be practiced after ensuring that piglets intake colostrum from their own mother (if possible) [134] and before the establishment of the teat order at the beginning of lactation [136]. However, sometimes this is not possible. Then, when piglets are fostered before ensuring a proper colostrum intake, one possibility could be to foster them on a sow with a similar stage in terms of colostrum production [129]; (3) lastly, another general recommendation would be that, whenever possible, make the minimum changes straight away [134].

Finally, the processing of piglets (normally practiced within the first two days after birth) is another management routine that could affect the piglets’ early performance. Such procedures include: an injectable administration of a compound with iron, an antiparasitic, and some other supplements. Tail docking and teeth clipping (not always) is also practiced and the piglets are identified with the farm number (ear tagging) [137]. The processing of piglets coincides with a vital period in which the piglets are still taking in colostrum and the suckling behavior is becoming established, which is very important for piglet development during lactation [138,139]. In addition, the processing of piglets (especially the more potentially painful procedures) may affect the piglets and results in pain a few days after the procedure [140] or even the possibility of missing nursing bouts [141]. These results, however, are not always consistent in the literature, as other authors have not observed these issues regarding missing nursing bouts, for example [142].

#### 3.1.5. Housing Interventions

The lactation period is crucial for the correct growth and development of pigs. During the lactation period, piglets are offered feed (creep-feeding) and water in order to facilitate habituation. This may help adaptation to solid feed and drinking water after weaning. Therefore, during the lactating period, the pigs’ feeding behavior is determined by milking and the beginning of solid feed intake. In addition to habituation, feeding behavior during lactation is influenced by two other aspects: social behavior (including with their littermates and the sow) and quality of the environment.

In natural conditions, pigs gradually start to interact with unfamiliar pigs around 12 d after birth [143]. This socialization phase helps them to develop their social skills and reduce aggressive interactions when dealing with unfamiliar pigs in the future [144]. In modern pig production, social interactions between pigs are infrequent during the suckling period. Early socialization, as referred to here, is the process of co-mingling piglets with unfamiliar conspecifics before weaning. A summary of early socialization effects are presented in Table 1. Figueroa et al. [145] determined the effects of social interactions between litters before weaning on maternal recognition. They studied how interactions between conspecifics before weaning may affect the pigs’ performance and behavior during the suckling period and after weaning by comparing early socialized litters (48 h after birth) with individual litters for the entire lactation period (28 days). It was observed that, during lactation, most early socialized pigs (94% ± 10%) suckled from their own mother. However, during the post-weaning period, the non-socialized piglets showed more aggressive interactions and a lower occurrence of positive social interactions. Moreover, the prevalence of severely wounded pigs was higher. Similar benefits of an early socialization were found in other studies [146,147,148], which included less aggressive behavior, accompanied by lower levels of stress after weaning. The effects on the growth of early socialization compared to single litters are less conclusive. Some studies found positive effects on growth [145,148,149,150], whereas others found no effects [146,151]. However, in any case, a negative effect of early socialization on piglet growth was never found.

Several studies [144,146,152,153] revealed that previous exposure to unfamiliar pigs during the suckling period could help to improve their social skills and reduce the length of the aggression period in intensive pig production. At weaning, this is of particular relevance as pigs establish new hierarchies, mostly based on agonistic behavior. However, when feeding behavior has been studied, socialization during lactation did not have any effect [149,152]. Although more research is necessary to confirm the effect of feeding behavior, it is likely that the improved performance at weaning mainly relies on a decrease in social stress and coping abilities of early socialized piglets [148]. In brief, allowing contact between litters before weaning may facilitate early interaction between conspecific animals reducing social stress during the post-weaning, resulting in better welfare and growth at weaning.

Social structure in gregarious species (i.e., pigs) may facilitate learning about what food types are safe to eat [154] and help pigs to increase their feed intake before weaning. Young mammals rely on older and more experienced conspecifics (i.e., their mother), for information when making the change from milk to solid feed [155]. Feeding behavior of pigs is mediated, among others, by social learning, such as observation of the sow. Currently, there are not many possibilities for social learning from the mother in farrowing crates, as sows are generally confined and cannot demonstrate the full range of explorative and feeding related behaviors. However, if the sow is kept in less confined systems (i.e., loose-housed), the sow’s behavior may direct the piglets’ behavior towards the feeder, modulated by social facilitation and stimulus enhancement [156]. Facilitating piglets to eat together with the sow in loose housing (Table 1) has been found to lead to a higher pre-weaning growth, most likely due to a higher solid feed intake [157]. This is probably due to the piglets having the opportunity to sample new foods, together with the sow, which reduces feed neophobia, increasing the intake of these new feeds compared to when the piglets feed separately from the sow [156,158].

The current housing facilities of lactating pigs in commercial farms are a rather barren environment where pigs have nothing else to interact with apart from the sow and their littermates. However, as reviewed by Vanheukelom et al. [159], increasing evidence suggests that pigs reared in enriched housing conditions express natural behavior (e.g., rooting), have less agonistic behavior and achieve better growth compared to pigs reared in barren housing conditions [153,160]. Providing environmental stimuli early in life may result in reduced reactivity to unfamiliar stimuli later in life and decreased feed neophobia, which can thereby enhance pre-weaning [161] and post-weaning feed intake [157].

Giving piglets an enriched environment before weaning may increase the development of foraging-related behaviors. Chewing, for example, is necessary to properly process solid feed, and providing a substrate that piglets can chew, may stimulate the development of muscles important for chewing, as well as the chewing behavior itself, which allows piglets to better adapt to ingesting solid feed [162]. In a study comparing outdoor vs. indoor housing for pigs (assuming that the outdoor system provides an enriched environment compared to indoor housing), the frequency of solid feed intake pre-weaning was higher compared to piglets reared indoors [163,164]. Piglets reared outdoors have a broader explorative, social, and feeding-related behavioral repertoire than indoor- reared piglets [165].

In addition, an enriched environment in the lactation pen increases the number of stimuli, enhances behavioral flexibility of pigs and reduces fear of novelty, improving the coping abilities of piglets to deal with stress at weaning [156,167].

### 3.2. Feeding Management

#### 3.2.1. Sows

During gestation, and specially in sows allocated in group, feeding management should foccus on providing enough feed to avoid abortion during the first third period [87], such as reducing hunger between meals with higher feeding levels and increased dietary fiber which can also promotes social stability and rest [87], or using foraging substrates [91].

Nutritional strategies and feeding management may have an impact on colostrum production and quality. Any positive feeding strategies to maximize colostrum production should involve mammary gland development and control colostrum synthesis in late-gestation (Figure 2). In order to maximize colostrum yield in practical conditions, overfeeding of sows during gestation should be avoided and it should be ensured that there is some degree of tissue mobilization at the end of gestation (although the optimum time in catabolic status before farrowing has not yet been described). However, a clear difference should be made for the first and second parity sows in terms of increasing the amount of feed offered during gestation. Then, in high productive sows, a flat curve during gestation is necessary in order to properly satisfy requirements for confirmed gestation (day 35 onward feeding 2.3 to 2.6 kg/day), allowing for the control of fat deposition until farrowing (±16 mm backfat), aiming for a balance between the control of overfeeding and piglet quality at birth.

In practical conditions, tissue mobilization at the end of gestation is strongly recommended for favoring maximum colostrum and milk production, and a balanced concept of this mobilization is required [89]. The practical approach is that 80% of the sows from day 108–110 of gestation are fed common lactation diets (high energy, crude protein (CP) and Lys content), while only a remaining 20% are fed the same gestating diets until farrowing. It is thought that feeding high-nutrient concentrated diets at late gestation (using lactation specification diets for the end of gestation) may not fulfill the concept of “balanced mobilization” before farrowing. However, for young sows that are still growing, the pattern of AA may allow mobilization without affecting growth.

A successful strategy for maximizing colostrum yield would be to keep sows at the end of gestation or from when they enter the farrowing crates with the same feeding level until farrowing, but shifting to a transition diet from 5–7 days before farrowing to 5–7 days after farrowing (Figure 2). The main remarks on feeding sows during gestation for maximum colostrum yield would be: (1) avoid overfeeding so that sows are not overfat; (2) promote the change from anabolic to a catabolic metabolic status of the sow in late gestation; and (3) follow the concept of “balanced catabolic status” before farrowing by shifting to a transition diet. This transition diet would include typical gestation energy content with lactation levels of protein and AAs.

Sows in gestation easily synthesize and deposit cover fat [170] and are restricted to avoid over fatness [171]; in fact, an excessive level of fat at the end of gestation, in addition to making delivery difficult, causes insulin resistance, metabolic imbalances and impairs subsequent lactation [171,172]. Therefore, the sow during gestation is usually restricted but is always overfed, assuming that the needs of this period are maintenance, the gravid uterus, and the mammary gland. In fact, one of the fundamental objectives of gestation is the recovery of body reserves in multiparous sows and that the gilts continue their growth.

In practice, feeding during gestation requires designing a feeding curve (kg of feed per sow per day). The shape of the curve depends both on the composition of the feed and on the temporal evolution of the nutritional recommendations. In practice, different feeding curves are used. In theory, one extreme would be trying to mimic the evolution of nutritional recommendations throughout the entire gestation according to those proposed by Solà-Oriol and Gasa [89]. This option is discarded as it is very expensive and logistically impractical. At the other extreme, a flat curve administering the same daily amount of feed without taking into account the temporal evolution of the recommendations is also used effectively in individually monitored sows (Figure 2).

Often two different feeds are proposed in an effort to make it easy for farmers and for feed-mill logistics: one “standard” feed, for multiparous sows, and one “special” feed, with higher nutrient content, intended for gilts and the rest of the sows for the last weeks of gestation. Table 2 shows the recommended values of standardized ileal digestible (SID) Lys and total Lys/kg of feed with 12.12 KJ according to the two sows categories based on prolificity hyper-prolific (HPr > 12–14 Total piglets born) and highly hyper-prolific (HHP > 14 Total piglets born; leaner animals). A SID Lys (g)/ME (Mcal) ratio of 1.6 is enough to satisfy the recommendations of all the HPr, except the young animals at the end of the gestation period (85–114 days) and the multiparous HHP for the period 0–85 days. The 1.9 ratio satisfies the need of the HPr gilts at the end of gestation and the HHP throughout the first two thirds of gestation (0–85 days). The ratio should be increased to 2.3 to satisfy the requirements of HHP sows during the last third of gestation. The content (%) of SID Lys and total Lys of the feed range between 0.45 and 0.65 and between 0.56 and 0.76, respectively. The recommendations for the other AA should be considered using the “Ideal Protein” concept reported in most nutrient requirement systems for swine, applied to the recommendations of SID Lys.

However, the main objective of the lactation period is to wean the maximum amount of kg of piglet without affecting the productive longevity of the sow. Lactation is the shortest period (15–19% of the total cycle), but may represent the period with the highest digestive and metabolic demand in the sow’s productive cycle. After farrowing, in a week or ten days, the sow increases the level of production by approximately 2.5 times, while feed intake does not follow the same evolution, then the sow is forced to mobilize body reserves. Excessive mobilization may seriously affect the sow’s reproductive future and reduces the sow’s longevity.

In this context, litter growth is directly related to milk production, which in turn depends on numerous factors; the same factors that balance the system by affecting both feed intake and body reserve mobilization. Among these factors, the genetic potential, the composition and handling of the feed (manual or automatically controlled), the environmental conditions and the interaction that is established between the sow and the litter, and even between both of these and the farmer stand out [89]. In our opinion, the composition of the feed and feeding management are not key factors of the production system, but they are the most important wildcard in the performance optimization and the economic return of the herd.

#### 3.2.2. Piglets

Complementary feed for suckling piglets, such as creep-feed or milk replacer might be useful due to the increment of number of piglets per sow. One of the major benefits is that it decreases the difference between the piglet’s energy requirements and nutrients obtained from milk, which increases as lactation advances.

Creep-feeds are highly palatable and easily digestible diets that are offered to lactating pigs after the first week or ten days of lactation. They are always formulated as complex diets but may vary by using highly palatable ingredients [173,174,175] combined with different technological processing. Creep-feeding is one of the most common earliest feeding strategies in solid feed to promote a suitable transition at weaning and may contribute to a reduction of pig BW variability from weaning onwards [89]. Creep-feed is especially beneficial for pigs raised in large litters with long lactations, as nowadays occurs with the hyper-prolific sows. Consumption of creep-feed during lactation stimulates feed intake and growth after weaning (higher average daily feed intake, ADFI and average daily gain, ADG; Figure 3) during the first days of post-weaning [176,177,178,179,180,181] and increases total BW gain [178,182] in eater pigs. Moreover, it has been reported that eaters of creep-feed may become familiarized early with a solid diet and they start to consume it early after weaning [176]. It is important to remark though that not all pigs within a litter consume creep-feed; approximately between 40% and 60% of pigs are actually creep-feed eaters [178,183,184].

A variation of creep-feed is using the same solid feed diet, but mixed with water or milk. The dry content of the liquid diet is similar to that of sow’s milk, satisfies nutrient and water requirements, and can solve the problem of learning to consume solid feed [185]. It has been observed that piglets fed the liquid creep-feed have faster growth [186] and higher villi after weaning [187]. However, Lawlor et al. [188] did not observe consistent effects of feeding a liquid diet during lactation.

Milk replacer supplementation is not as usual as creep-feed, but it may have similar consequences. Milk replacer can be used differently: (1) moving the surplus piglets out of the farrowing pen when they are a few days old and giving them milk replacer, although this has been associated with reduced welfare for both piglets and sows [189], or (2) keeping all piglets with the sow and supplying the milk replacer continuously throughout lactation. It has been observed that providing milk replacer during lactation increases piglets’ weaning BW and total litter BW [190,191,192,193], gain weight after weaning [180], as well as piglet survival [189]. However, Douglas et al. [14] did not observe an improvement in performance, but rather a reduction in weight variation in LBW piglets. Supplementing piglets with milk replacer has no effect on sows’ feed intake, back fat or milk production [191,192,193,194]. In addition, milk replacer supplementation may help piglets to become familiar with solid feed after weaning according to gradually replacing milk formula by solid feed as a result of the feeding strategy used. A clear seasonality has been reported during the warmer months. Pigs supplemented with milk replacer consumed more milk, and therefore had greater BW at weaning compared to pigs without milk replacer supplementation [194,195,196]. This can be explained because sows during warm months suffer heat stress, which decreases the sows’ feed intake and consequently reduces milk production [195].

Overall, creep-feeding and milk replacer supplementation can be an effective strategy for reducing BW variation at weaning [192], as well as being a tool for those piglets with growth retardation.

### 3.3. Weaning

Weaning in intensive pig production, usually occurring between day 21 and 28 of age and it is when piglets are separated from their mothers and taken to a new environment, mixed with other piglets, and switched from a liquid diet (20% DM) to a compound dry feed [197]. The nutritional, psychological, and environmental changes produce a stress response and anorexia on the first days after weaning [198,199]. Anorexia and stress produce gastrointestinal disturbances: alterations in the small intestine architecture and enzyme activities, transiently-increased mucosal permeability, disturbed absorptive-secretory electrolyte balance and altered local inflammatory cytokine patterns [200]. These can cause diarrhea, sub-optimal growth, and (or) increased morbidity and (or) mortality [199].

As we described earlier, the early events occurring at a very young age, even before the pigs are born, might negatively affect the performance results of the nursery, growing, and finishing pigs. Therefore, all the previous management strategies, like sow housing and feeding, fostering, ensuring colostrum and milk intake, early socialization, using complementary feed during lactation, etc., may help to improve the post-weaning growth check. However, not only management strategies are required, but also, in the following section, a set of dietary strategies will be reviewed.

## 4. Nutritional Interventions

### 4.1. Sow

#### 4.1.1. Enhancing Fetal Growth

An early nutritional intervention during gestation to reduce the incidence of IUGR and LBW pigs is to provide functional nutritional strategies at two different levels during gestation: (1) between day 12 and 25 after mating to reduce the peak of embryonic death and implantation failure, and (2) from day 35 to 75 of gestation when fetal losses occur due to the inadequate development of the placenta or insufficient uterine capacity that impairs the proper development and nutrient uptake of the fetus [201,202]. During the last two thirds of gestation, a high decrease in non-essential AA is observed (Branched Chain AAs (BCAA) in the Krebs cycle and Arg and Glu in the Urea cycle), becoming non-available for their key metabolic pathways [89,203,204].

For the first third of gestation, and due to the high demand of AA in the uterine lumen around the peri-implantation period, the AA uptake provided by the sow’s diet play an important role for protein synthesis and activation of cellular functions in the intrauterine environment (with particular emphasis on Arg, Leu and Gln). Moreover, the histotrophic process occurring during the implantation is related to a pro-inflammatory process with a high intervention of cytokines, limphokines, hormones, enzymes and growth factors that may be recognized as hazardous in relation to a hyperactive intrauterine mucosa [205,206]. Therefore, diets and functional nutrients or compounds, provided with the aim of reducing the inflammatory process and the oxidative stress, may be positive for avoiding early reabsorptions.

During the second third and in late gestation, the sow is focused on increasing the placenta function. The main objective for the second third of gestation is providing enough substrates and nutrients to regulate gene expression, protein synthesis, and angiogenesis in order to maximize placenta development. Dietary supplementation of certain nutrients may enhance litter size and fetal growth, such as chromium, L-carnitine, omega fatty acid, Lys, and L-Arg. Moreover, increasing levels of BCAA at the end of gestation, linking with transition diets, may help to satisfy a “balanced catabolic process” at the end of the gestation period. In addition, piglets born from sows supplemented with these nutrients are more vigorous, and able to suckle for longer; therefore, they receive more milk from the sow and grow faster during the suckling period [207].

Table 3 shows a summary of different nutrients supplemented during gestation that may enhance fetal growth. Among these nutrients, Arginine supplementation during early gestation may improve placental weight, prolificity, and growth of the fetus [208,209,210], however L-Arg supplementation at late gestation (>day 80) does not have a clear effect on prolificity, BW of the fetus, or further piglet performance [211]. This is because Arg is utilized by multiple pathways, including the synthesis of protein, nitric oxide (NO), polyamines, and creatinine [212]. Polyamines and NO are key regulators of angiogenesis, embryogenesis, and placental and fetal growth, mainly at the beginning of pregnancy [212]. Therefore, L-Arg may promote optimal intrauterine conditions for minimizing losses and improving growth of viable fetuses in early gestation. However, the optimum timing and duration of L-Arg supplementation still needs to be established.

Maternal diet enriched with methylated micronutrients, such as folate, betaine, or vitamin B_12_ has also been associated with increased fetal weight in late gestation [213]. These compounds provide beneficial effects for embryo and placental development, and can increase litter size in multiparous sows [214].

Supplementing dams with L-Carnitine results in a larger placenta, heavier piglets at birth [215,216,217], greater muscular area, more total muscle fibers in piglets [218], and increased circulating concentrations of insulin-growth factor I (IGF-I) at mid-gestation in sows [215], which stimulates the proliferation and differentiation of skeletal muscle cells and regulates muscle growth and development.

Chromium, which potentiates the action of insulin [219], if supplemented throughout gestation, increases the sow’s body mass gain [220], farrowing rate [221], the total number of piglets born alive [220,221], and produces a progeny with a higher number of muscle fiber at birth, weaning, and slaughter [222]. However, Zinc, which plays a key role in protein synthesis, nucleic acid metabolism, and immune function [223], if supplemented during the entire gestation period, increases birth weight and the number of pigs weaned per litter [224]; while during the last third, it has a positive effect on intestinal development and the immune function of the offspring [225]. However, a deficiency in Zn during gestation can result in abnormalities in fetal or skeletal growth, general body growth retardation, or dermatitis [226].

#### 4.1.2. Increasing Colostrum and Milk Production

Mammary gland growth, which occurs during gestation and lactation, is critical for lactogenesis. However, the number and vitality of suckling piglets, and the total amount of feed consumed by the sow and the shape of the feeding curve during lactation are the main factors that condition milk production. Kim et al. [171] proposed that, to improve milk yield performance, it is necessary to increase mammary gland growth (including vascular growth) and blood flow to mammary tissue. Arginine is the common substrate for the generation of NO (a major vasodilator and angiogenic factor) and polyamines (key regulators of protein synthesis). Therefore, a modulation of the Arg-NO pathway may be a new strategy to improve the lactation performance by enhancing the development of the mammary gland and its uptake of nutrients. In support of this proposition, Mateo et al. [227] observed increased milk production and growth of piglets when primiparous lactating sows where supplemented with 0.83% L-Arg (as 1% L-Arg-HCl).

Branched chain AAs are spared by the liver and readily used in tissues with high metabolism. Catabolism of BCAA, such as valine, occurs in the liver, kidney, muscle, heart, adipose tissue, and mammary gland. Research with lactating sows has indicated that valine is catabolized at a high rate in the mammary tissue [228] and it is contained in the composition of milk AAs at different ratios (Table 4, adapted from different sources [229,230,231]).

Several research projects have been conducted in the last decades to estimate the correct dietary Val-to-Lys ratio (Val:Lys) for lactating sows in order to maximize litter performance and minimize tissue mobilization during lactation. The present standard ileal digestible (SID) Val:Lys recommendations vary between 70% and 85% for lactating sows; however, experimental trials using 120% Val:Lys observed positive effects on the number of pigs weaned per litter [232,233,234,235,236]. In contrast, other studies did not confirm this high value [237,238,239]. A meta-analysis of the available data on lactating sows based on the effect of the Val:Lys ratio on the litter weight gain confirmed the 0.90:1 Val:Lys ratio recommendation [233,234,235,238,239,240]. The current recommendation for SID Val:Lys is 0.85:1 [241], whereas the Danish recommendation of SID Val:Lys is 0.76:1 (total Val:Lys of 0.80:1; [242]). Therefore, for modern high production sows the Val to Lys ratio, which is important to take into account when high performance is expected.

L-Carnitine is required for the transport of medium-chain fatty and long-chain fatty acids into the mitochondria for β-oxidation. Ramanau et al. [243] described that sows whose diet is supplemented with L-carnitine produce more milk during lactation than control sows, even in a strongly negative energy and protein balance. The study indicates that L-carnitine might also enhance the utilization of body fat by sows in a strongly negative energy balance, particularly in primiparous sows.

Herbal and pharmaceutical galactagogues are used in humans to enhance milk production [244,245]. The most frequently used products include fenugreek, galega, and Mary’s thistle. Fenugreek, which is a seed belonging to the pea family, has several medicinal properties, including galactagogue effects [246]. For example, fenugreek contains chemical compounds such as flavonoids, terpenoids, and saponins (diosgenin), which are known phytoestrogens that enhance milk production by stimulating the anterior pituitary gland to increase prolactin [247,248]. Other compounds, such as silymarin (active extract of milk thistle) may also have a galactagogic effect. It contains flavonolignans, bioflavonoid phytoestrogens with a steroid-like structure, which might explain their ability to stabilize plasma membranes and protect the liver by promoting detoxification [249]. It is also possible that they could act on estrogen receptors by limiting the endogenous receptor antagonism of milk production [250]. Sylimarin increased prolactin levels in female rats [251] and sows [252]. Ginger, a spice that is believed to increase blood circulation and consequently vasodilatation, may also improve milk production [248]. It has also been observed that adding phytogenic actives (PA) to the sow’s diet during late gestation increases IgG in milk [253], and during gestation and lactation, and can increase protein content in colostrum and fat content in milk [254]. The authors also observed that milk had an inhibitory activity against Bacillus subtilis and Staphylococcus aureus [254]. These findings open the door to new possibilities for improving colostrum and milk quality.

#### 4.1.3. Flavor Transference through Amniotic Fluid and Milk

Sows, like most mammals, have an innate recognition and preference for high energy (sweet savors), proteins (umami savor), and even electrolytes (savory savors) in feeds [255,256], which make sows capable of selecting and ingesting an appropriate diet. The rest of the flavors are identified as a challenge. Neonate pigs can differentiate auditory, olfactory, visual, and tactile stimuli after birth [257]. They are also able to recognize their mother’s fecal and skin odors [258] and amniotic fluid [259], which help them recognize their mother, localize the nipple, and initiate suckling [260]. Therefore, if mothers know what kind of food is good to eat and what kind of food is present in their environment, maternal learning would mean that offspring would prefer to eat the same. Maternal learning is a feeding behavior that facilitates the search for food, making it more efficient and adaptive, and it is established during gestation and lactation [261].

It has been reported that some volatile compounds can cross the placental barrier, enter the fetal bloodstream, and diffuse out of the nasal blood capillaries and contact the fetal olfactory receptors. In addition, flavors diffuse into the amniotic fluid where the fetus inhales or swallows it which stimulate olfactory receptors or taste buds [262]. As a result, the exposure to flavors may lead to creating a preference to those flavors later in life and can positively modulate the acceptance of food with similar flavors before and after weaning [261]. After birth, maternal learning could continue through the milk, the hedonism and lactic postingestive effect and the pleasure of nursing may create associative learning with cues in milk [263]. Thus, the flavor preference might be strengthened more when the flavor is also present in the maternal milk [261,264]. However, just milk exposure does not increase flavor preference in all species [263]; in pigs, postnatal exposure alone did not ameliorate the post-weaning associated problems (low feed intake and gain, diarrhea, and stress) [265].

There are several compounds that have been detected in amniotic fluid, colostrum, and milk in different species. In sows the compounds are anethol, borneol, cinnamaldehyde, eucalyptus, eugenol, limonene, linalool, p-cymene, and thymol in the amniotic fluid [254,266,267], and α-pinene, anethol, carvone, cinnamaldehyde, eugenol, limonene, menthol, and thymol in milk [254,266,267,268,269,270,271]. The supplementation of one or more of these compounds during gestation and/or lactation can have a positive impact on the feeding behavior of neonates and post-weaning pigs. Table 5 shows a summary of different studies on this topic.

Therefore, maternal learning has positive effects on pigs feed intake, growth, and feeding behavior, and can reduce post-weaning stress. However, not only the flavor supplementation during gestation and lactation can change the organoleptic composition of the maternal fluids (amniotic fluid and milk), it can also modify the chemical composition of colostrum and milk and it might alter the microbiota. Consequently, maternal transference (which includes maternal learning) is an indirect strategy through the sow’s diet for modulating pigs’ feeding behavior and growth, milk and colostrum composition, and even the bacteriostatic activity of milk.

#### 4.1.4. Modulating Antioxidant and Inflammatory Responses

Reactive oxygen species levels and oxidation, either in sows or piglets, play a relevant role for the growth of piglets during the lactation and post-weaning period, which opens up opportunities to guide nutrient interventions to alleviate oxidative damage. A beneficial antioxidant status in sows could attenuate oxidative stress-related long-lasting effects on the offspring [274,275]. In this respect, vitamins, trace minerals, and biological antioxidants may play important roles in scavenging the oxygen free radicals. Different results have recently been published on the effect of different dietary interventions in sow diets in relation to litter performance. Only a few studies have reported long lasting effects after weaning (Table 6), which raises interest in performing these kinds of studies.

Omega-3 (n-3) polyunsaturated fatty acids (PUFA) are precursors of lipid mediators that play an important role in the regulation of inflammation through the production of eicosanoids [276]. Marine fish are the main sources of n-3 fatty acids (EPA and DHA) [277]. It has been observed that EPA and DHA can alleviate and prevent the development of inflammatory processes in humans [276]. The addition of PUFA in the sow’s gestation and/or lactation diets can increase the BW [278,279] of piglets, enrich the neonatal piglets with n-3 PUFA [278,280], and reduce the post-weaning inflammation process [279].

Vitamin E and Selenium are the most well-known biological antioxidants. Prenatal vitamin E and Se supplementation through the dam can provide an effective antioxidant status of the piglet at birth, while postnatal supplementation may be the main determinant of progeny antioxidant status during the lactation period and after weaning. The addition of vitamin E and/or Se in the sow’s diet during gestation and lactation improved the number of piglets born and weaned, the weight of piglets at weaning, and enhanced milk fat content, humoral immune function (IgG and IgA) and the antioxidant activity in sows and piglets [281,282,283]. The effects were higher when Se was provided in organic sources (Se yeast or selenomethionine) compared to sodium selenite [284,285].

However, plant extracts may be rich sources of antioxidant phenolic and polyphenolic compounds. Phytogenic actives are a source of various bioactive compound groups, such as terpenes, phenols, glycosides, saccharides, aldehydes, esters, and alcohols. Some described effects are stimulation of digestive secretions, immune stimulation and anti-inflammatory activities, intestinal microflora modulation, and antioxidant effects [286,287], as well as estrogenic and hyperprolactinaemic properties [288,289]. Hossain et al. [290] compared the antioxidant activity of 30 common spices, and highlighted the high activity of rosmarinic acid from *Lamiaceae*, eugenol from clove, kaempferol from *Apiaceae* spice, curcumin from turmeric, capsaicin from chili, gingerol from ginger, and thymol from thyme.

A literature review describes a number of studies with different phytogenic compounds. Recently, a blend of PA supplemented during gestation and lactation [254], increased blood serum catalase activity and NO levels at early gestation (day 35), superoxide dismutase (SOD) and glutathione peroxidase (GSH-Px) at d 110 of gestation, and greater litter size at farrowing. A variety of compounds, such as resveratrol, grape seed polyphenols, oregano essential oil, garcinol, alpha-lipoic, seaweed extract, and hemp seed extract may promote positive effects in the antioxidant activity in sows and their litter (Table 6), which has been associated with an improved composition of colostrum and milk (higher levels of lactose, fat, IgG, etc.), and increased BW and piglet survival rate at birth [52,254,291,292,293,294,295,296,297].

Another product that can modulate the immune response of pigs is spray dried plasma (SDP) [298,299,300], which is rich in highly digestible protein [301] and is obtained from the industrial fractionation of healthy porcine or cattle blood. Several studies support the use of SDP as a protein source in nurseries as it improves piglet performance and reduces the incidence of post-weaning diarrhea [302], especially during the first two weeks after weaning and in farms with lower sanitary conditions [303], reduce intestinal inflammation, and maintain gut integrity [298,299,300]. The pig’s improved health status could be explained because spray drying makes it possible to preserve the immunoglobulins, growth factors, bioactive peptides, and other biological components present in the blood that can interact with the gut-associated lymphoid tissue (GALT) [304,305]. Other studies have found that SDP can also positively affect the immune response of the bronchoalveolar-associated lymphoid tissue [306] or the genito-urinary-associated lymphoid tissue [307] since GALT is interconnected to others by the common mucosal immune system [304]. Few studies have examined the use of SDP in sows, but the results show that it has a positive effect on productive and reproductive performance. When 0.5% of SDP was added to the lactation diet, it was found that there was an increase in individual pig weight at weaning [308,309], an improved survival rate during lactation [310,311], an increased subsequent parturition rate, and a decrease in the wean to first estrus interval for primiparous sows [308]. Even when 0.5% of SDP was only included in the gestation diet an increase in pig weight at weaning was observed [312]. In addition, porcine reproductive and respiratory syndrome in unstable farms showed an increase in the parturition rate and the number of piglets born alive and weaned per litter [313]. These findings, presented in Table 6, suggest that SDP is a potential ingredient for modulating the sow’s systemic inflammatory response, and thereby improving the sow’s performance.

#### 4.1.5. Modulating Microbiota with Probiotics

The possibility of an entero-mammary route for microbial transfer, together with the natural exposition of the piglets to the sow’s feces in the nursery, opens the possibility of gut microbiota modulation in pigs by supplementing sows with probiotics. A summary of their effects is presented in Table 7. A recent work shows how the treatment of pregnant sows with a novel mixed probiotic culture, resulted in an improved microbiota diversity in the neonatal pigs [315]. A study by Starke et al. [316] also showed that supplementing an *E. faecium* strain to sows during the month before labor, modified the fecal microbiota of the mother and also from the piglets during the lactation and post-weaning period, although changes did no not mirror the quantitative changes on sows. The mothers’ responses were, however, variable and this was suggested to be dependent on the previous bacterial composition of their gut microbiota. In this regard an in-field trial providing a probiotic based on *B. subtilis* C-3102 during gestation and lactation, and to progeny after weaning, showed an impact on the microbiota of the mothers and their progeny, showing that pigs born from probiotic-fed sows had a similar fecal microbial population as their mothers [317]. The use of probiotics could also play a role in preventing IUGR pigs. In this regard, it has been described how supplementing weaning diets with *Bacillus amyloliquefaciens* can decrease the inflammatory response and regulate small intestinal microbiota of IUGR weanlings pigs [318].

Barba-Vidal et al. [319] extensively reviewed the use of probiotics in reproductive sows to improve performance. Numerous studies (Table 7) can be found in the literature that assess the effects of different probiotic strains when administered to sows, with positive effects on the performance of pigs, increasing growth rates and reducing clinical signs of post-weaning diarrhea. These improvements could be related to the aforementioned improved microbiota balance, facilitated by the probiotic-treated mother, but also to increases in feed intake, reported by several authors, due to the use of probiotics in pregnant and lactating sows [320,321,322]. Increased feed intake is also expected to improve the body condition score of the dams at the end of lactation and reduce weaning-estrus interval [320,322,323]. Higher intakes and lower weight losses would also increase milk production [324,325] and therefore enhance litter size [326,327]. Tsukahara et al. [328] demonstrated that probiotic supplementation improved their reproductive performance of reproductive sows challenged with porcine epidemic diarrhea virus, with a higher BW one week post-partum and greater milk production. Probiotic treated sows also tended to return to estrus faster and had heavier piglets at birth with a lower mortality percentage during early days of suckling.

Probiotics, therefore, are a potential tool for improving the performance of sows and their litters, although significant benefits are not always found. The effects and potential of probiotic strategies will depend on the properties of the probiotic strain and also on the gut ecosystem into which it is introduced [319]. While the evidence of the potential of probiotics in sows is nowadays clear, the action mechanisms involved are still not known. Changes promoted in the gut microbiota appears to be the most plausible mechanisms that could turn the sow into an excellent microorganism donor to the piglet, facilitating a beneficial gut colonization and offering better training to their immune system. However, probiotics could also exert effects on the mother’s metabolism, with implications for her ability to minimize weight losses during lactation and increase milk production, resulting in improved breeding success.

### 4.2. Piglets

The nutritional support of gastrointestinal growth and function is an important consideration in the clinical care of neonatal animals. The literature describes specific nutrient (e.g., glutamate, glutamine, tryptophan, nucleotides, etc.) requirements for the infant gut, which are summarized in Table 8.

Nucleotides, a combination of a nitrogenous base, a pentose sugar, and a phosphate group [339], have been suggested to be “conditionally essential” nutrients during a period of rapid growth, stress, injury, and immunodeficiency [339,340], such us early weaning [341]. These processes are heavily dependent on availability of DNA, RNA, and ATP energy, whose synthesis depends on availability of nucleotides. Therefore, lactating pigs and weanling pigs require nucleotides. Supplementing nucleotides in weanling pigs improves intestinal function, immunity, health, nutrient utilization, and preserves energy [3,339,342,343].

Targeted Amino-acids due to their functional role could be supplied during a certain period depending on the animal status. For example, inflammation increases tryptophan (Trp) catabolism and may thus decrease Trp availability for growth [344]. In addition, weaned pigs in challenging conditions have lower plasma levels of Trp than pigs in healthy conditions [345]. Consequently, providing of synthetic Trp, above the requirement, increases availability for growth [346,347]. In addition, some amino acids, especially Leucine, control some components of the protein synthesis [348]. Pulsatile delivery of leucine has been documented to be an effective strategy for increasing protein synthesis and the lean growth of neonatal piglets [349,350,351]. When Leu is supplemented in low-protein diets, it increases protein synthesis in muscles and organs, and also increases daily weight gain [352]. Glutamic acid (Glu) and glutamine (Gln) are not usually considered to be essential nutrients; however, providing of supplemental Gln to both suckling and weaned pigs has shown improvements in growth and health, most probably related to improved intestinal status and immune function [353,354]. The addition of Glu has even been reported to enhance intestinal-mucosal mass and barrier function (tight junctions), and to influence the expression of AA receptors and transporters in the jejunum of weaning piglets [355], which is beneficial for improving digestion and absorption.

Therefore, to improve piglets’ growth and development after weaning, we need to implement the previous dietary strategies, without forgetting some nutrients that become essential to piglets in early growth. A strategy to include these nutrients (nucleotides, tryptophan, glutamic acid, etc.) would be to supplement them in the creep-feed or in the milk replacer. Overall, dietary strategies are as important as management ones.

## 5. Conclusions

After reviewing the literature, it appears evident that there are several solutions for improving the performance of young pigs. The present review evidenced the need for an integral approach based on management, dietary, and feeding strategies to improve the pig’s performance during the first days of life. Management strategies would include adequate housing during gestation, reducing pain at farrowing, ensuring colostrum intake, promoting early socialization, and habituation to solid feed. Dietary strategies should consider tuning feeding of sows during gestation (to enhance fetal growth and vitality of piglets) and lactation (to increase colostrum and milk production), and feeding piglets with specific nutrients during the first days of life to enhance an early feed intake, adequate microbiota, and healthy gut. Therefore, a combination of different strategies is essential to produce robust pigs and to achieve the highest performance potential throughout their entire life.

## Figures and Tables

**Figure 1 animals-11-00302-f001:**
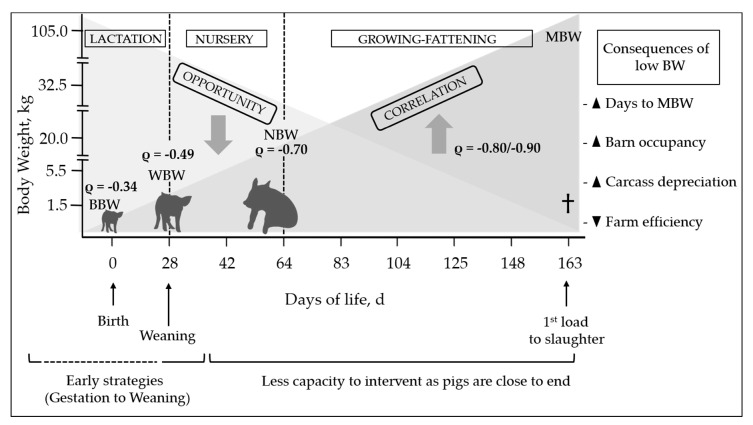
Effect of birth and weaning body weight until slaughter. The spearman correlation coefficient (ρ) indicates the correlation between body weights at different production phases with the number of days to reach a market body weight of 105 kg. BBW: birth body weight; WBW: weaning body weight; NBW: nursery body weight; MBM: market body weight. † means slaughter house [2,7].

**Figure 2 animals-11-00302-f002:**
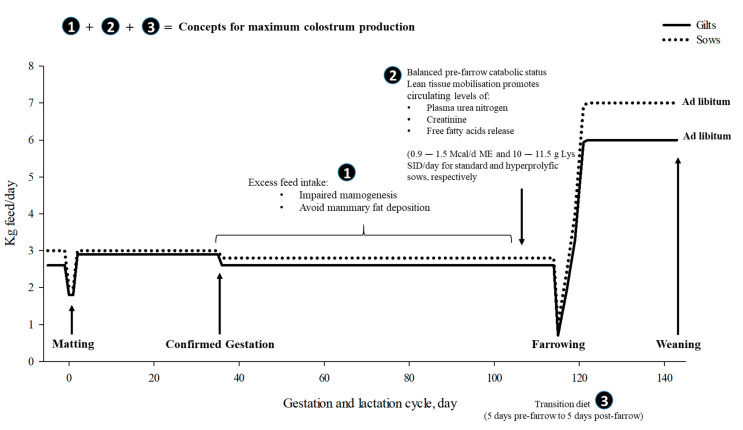
Nutritional strategies and feeding management concepts for maximum colostrum production in gilts and sows [89,168,169].

**Figure 3 animals-11-00302-f003:**
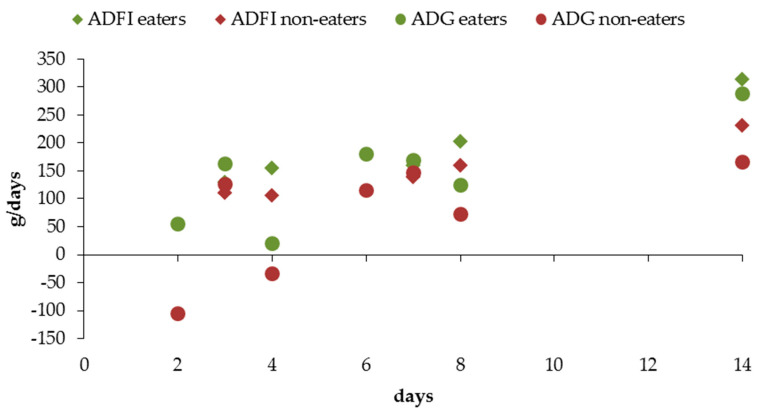
Post-weaning feed intake (average daily feed intake, ADFI, ♦) and growth (average daily gain, ADG, ●) of piglets categorized as creep-feed eaters (eaters, green), and non-creep-feed eaters (non-eaters, red) [176,177,178,180,181].

**Table 1 animals-11-00302-t001:** Performance and welfare-associated effects before and after weaning of different housing interventions during lactation in piglets.

Housing Strategy	Performance		Welfare		Reference
	Pre-Weaning	Post-Weaning	Pre-Weaning	Post-Weaning	
Environmental enrichment	-	-	↑ play behavior	↓ bite injuries	[153]
	↑ ADG, ≈Feed intake	↑ Feed intake	-	-	[157]
	≈ADG	≈ADG	↑ play behavior	↓ stress biomarker (cortisol)	[166]
Early socialization	-	↑ ADG	-	↓ n° aggressions, ↓ positive social behavior	[145]
	≈ADG	↑ ADG	-	↓aggressive behavior	[149]
	≈ADG	↑ BW ↑ ADG	≈bite injuries	↓ bite injuries ↓ intensive aggressive behavior	[150]
	-	-	↑ bite injuries	↓ bite injuries, ↓ stress biomarker (cortisol, α-amylase, and chromogranin)	[148]
	≈ADG		↑ bite injuries	↓ bite injuries, ≈stress biomarker (cortisol)	[146]
	-	-	-	↓ n° aggressions, shorter aggressions	[147]
Loose housing	↑ ADG, ≈Feed intake	-	-	-	[157]
	-	-	-	↓ damaging behavior (belly nosing) ↓ manipulative behavior, ↑ chewing ↑ feed exploration, ↑ play behavior	[156]
	-	↑ ADG ↑ Feed intake	-	↑ feed exploration ↓ damaging behavior (tail biting) ↓ signs of diarrhea	[158]

ADG = average daily gain; BW = body weight; (↑) = increase; (↓) = decrease; (≈) = no change; n° = number.

**Table 2 animals-11-00302-t002:** Recommended SID Lys/ME ratios and levels of SID Lys and Total Lys (g/kg feed) for the present hyper-prolific and high hyper-prolific gilts and multiparous sows between day 0 to 85 and 85 to 114 of gestation.

SID Lys (g)/ME (MJ)	Hyper-Prolific				High Hyper-Prolific			
	Gilts		Multiparous		Gilts		Multiparous	
	0–85 Days	85–114 Days	0–85 Days	85–114 Days	0–85 Days	85–114 Days	0–85 Days	85–114 Days
0.38	X	-	X	X	-	-	X	-
0.45	-	X	-	-	X	-	-	-
0.55	-	-	-	-	-	X	-	X
SID Lys (g/kg feed) ^1^	4.5	5.5	4.5	4.5	5.5	6.5	4.5	6.5
Total Lys (g/kg feed)	5.6	6.6	5.6	5.6	6.6	7.6	5.6	7.6

^1^ Feed energy content: 12.12 MJ ME/kg.

**Table 3 animals-11-00302-t003:** Effects of early nutritional intervention during gestation to enhance fetal growth.

Nutrient	Supplementation	Product	Inclusion Level	Sows’ Performance	Piglets’ Performance	Reference
Arginine	days 30 to farrow	L-Arg	1%	↑ placental weight, ↑ angiogenesis ↑ Arg and insulin in sows plasma	↑ piglet BW	[208]
	days 14 to 28	L-Arg	25 g/day	↑ prolificity, ↓ fetal reabsorption ↑ growth of viable fetus	-	[209]
	days 14 to 28	L-Arg	26 g/day	↑ prolificity, ↑ growth fetus ↑ myofiber formation in fetus	-	[210]
	days 93 to farrow	L-Arg	26.8 g/day	=prolificity, =maternal IGF-I, insulin, and urea nitrogen	=piglet BW -	[211]
	days 81 to farrow	L-Arg	27.6 g/day	=litter and sow performance		[211]
Methylating micronutrients	10 days before insemination to day 91	Methionine Choline Folic acid Vitamin B_6_ Vitamin B_12_ Zinc	4700 mg/kg 2230 mg/kg 92.2 mg/kg 1180 mg/kg 5930 µg/kg 149 mg/kg	↑ prolificity; ↑ fetal weight ↑ embryo and placental development ↑ IGF-II in fetal muscle ↑ Met metabolism donating monocarbon units for the remethylation of Met from the toxic homocysteine	-	[213]
L-Carnitine	days 5 to 112	L-carnitine	100 mg/day	↑ sow BW gain and fat ↑ litter birth weight ↑ number of pigs born alive ↑ IGF-I d 60 and 90	↑ litter weaning weight	[215]
	days 1 to farrowing	L-carnitine	125 mg/day	↑ sow BW gain, ↑ litter birth weight ↓ non-viable pigs	-	[216,217]
	days 1 to farrowing	L-carnitine	50 mg/kg	=pig birth weight; =IGF-I	↑ muscular area and muscle fiber number	[218]
Chromium	all gestation	Cr picolinate	200 µg/kg	↑ litter size and weight at birth ↑ efficiency of insulin action (↓ insulin pre and post-feeding and ↓ insulin:glucose ratio)	-	[219]
	all gestation	Cr picolinate	400 µg/kg	↑ sow and litter body gain mass ↑ pigs born alive, ↑ Cr in colostrum and serum during gestation ↓ serum insulin, glucose and serum urea nitrogen	↑ pigs/litter at weaning	[220]
	all gestation	Cr picolinate	200 µg/kg	↑ total pigs born and born alive	-	[221]
	all gestation	Cr picolinate	400 µg/kg	=litter performance	↑ number of muscle fiber at birth, weaning, and slaughter	[222]
Zinc	days 15 to farrowing	Zn AA	100 mg/kg	↑ pigs born and weaned per litter	-	[224]
	last third	Zn AA	250 mg/kg	↑ number of live pigs	↑ Zn serum day 7 and weaning ↑ VH and VH:CD	[225]

Arg = arginine; BW = body weight; Cr = chromium; IGF = insulin-growth factor; Met = methionine; VH = villus height; VH:CD: villus height and creep depth ratio; Zn = zinc; (↑) = increase; (↓)= decrease; (≈) = no change.

**Table 4 animals-11-00302-t004:** Pattern of the different AAs ratios to lysine for different pools in lactating sows (adapted from different sources [229,230,231]).

AAs	Milk	Mammary Tissue	Retained in Mammary Gland	Plasma
Lysine	100	100	100	100
Threonine	59	58	150	86
Tryptophan	77	78	25	69
Methionine	114	116	53	32
Valine	60	58	216	166
Leucine	56	58	403	164
Isoleucine	65	89	231	72

**Table 5 animals-11-00302-t005:** Effects of prenatal and postnatal exposure of different flavors or compounds present in the sow’s feed.

Flavor/Compound	Dose	Supplementation Time	Post-Weaning Effects	Reference
Fiannor number	50 mg/kg	3 weeks during L	↑ feed intake with the flavor ↑ BW	[272]
Anethol	350 mg/day	days 98 to 108 of G	↑ feed intake with the flavor ↑ BW ↓ diarrhea ↓ post-weaning stress	[264,265]
Cheese and anethol	1.5 g/kg 0.75 g/kg	Last 2 weeks of G	↑ preference	[259]
Anethol, cinnamaldehyde, and eugenol	375 mg/kg	Last 40 days of G and all L	↑ feed intake, BW, and ADG	[266]
Limonene and cinnamaldehyde	0.1%	Last 40 days of G and all L	↑ feed intake, BW, and ADG	[267]
Menthol, carvone, and anethol	0.1%	Last 40 days of G and all L	↑ feed intake, BW, and ADG	[267]
Monosodium glutamate	5%	Last 30 days of G and all L	↑ sucrose intake ↓ preference for MSG	[273]

ADG = average daily gain; BW = body weight; G = gestation; L = lactation; MSG = monosodium glutamate; (↑) = increase; (↓)= decrease; (≈) = no change.

**Table 6 animals-11-00302-t006:** Effects of fatty acids, vitamins, trace minerals, phytogenic compounds, and spray-dried plasma in the sow diet on the oxidation and performance of sows and piglets.

Ingredient	Diet	Sows’ Performance and Oxidation	Suckling Piglets	Post-Weaning Period	Reference
n-3 PUFA	G (tuna oil)	-	↑ BW, ↑ n-3 fatty acids	↑ BW	[278]
	late G and L (protected fish oil)	-	-	↑ BW, ↓ cortisol, ↓ haptoglobin ↓ IL-1β, IL-6, and TNF-α	[279]
	G (menhaden fish oil)	↑ n-3 PUFA in serum and milk ↓ arachidonic acid in serum and milk	↑ n-3 PUFA and eicosapentaenoic acid levels in serum	-	[280]
Vitamin E	G and L (α-tocopherol acetate, 500 mg/kg + vitamin C, 10 g/day)	=IgG and IgA in colostrum and milk	↑ total Ig and IgG	-	[282]
	last week of G and L (vitamin E, 250 IU/g)	↑ IgG, IgA in plasma, colostrum, and milk ↑ milk fat content	↑ weaning BW, ↑IgG and IgA ↑ total antioxidant capacity ↑ CAT	-	[281]
	G (α-tocopherol: 50 mg/kg; selenium (Na_2_SeO_3_) 30 mg on days 30, 60 and 90)	↑ piglets born/litter, ↑ weaned piglets/litter	↑ weaning BW, ↑ IgG	-	[283]
Selenium	late G and L (0.3 vs. 1.2 mg/kg Se as Se-yeast)	↑ protein, lactose, solids-not-fat, Se, and IgM in colostrum ↑ fat and IgA in milk	↑ GSH-Px activity, ↓ MDA content, ↑ IgA and IgG ↓ mortality	-	[284]
	G and L (+0.3 mg Se/kg sodium selenite vs. selenomethionine)	-	Selenomethionine = ↑ weaning BW, ↑ GSH-Px, SOD activity, ↓ MDA content; ↑ digestive enzymes of protease, amylase, and lipase	-	[285]
Phytogenic compounds	G and L (resveratrol, 300 mg/kg)	↑ lactose in colostrum ↑ milk fat	↑ plasma HDL and LDL ↑ enzyme activity and mRNA level related to lipolysis, fatty acid uptake from circulating triacylglycerols and lipogenesis	-	[291]
	late G and L (grape seed polyphenols, 200 or 300 mg/kg)	↑ GSH-Px and SOD activity ↑ progesterone and estradiol ↑ farrowing survival ↑ IgM and IgG in colostrum	↓ mortality	-	[292]
	G and L (blend of phytogenic actives, 1 g/kg)	↑ GSH-Px and SOD activity ↑ farrowing survival ↑ protein in colostrum and ↑ milk fat	↑ CAT and SOD activity ↑ MUC2, digestive enzyme IDO ↑ immune response PPARGC-α, TNF-α, TGF-β1, and IL-10 genes	↑ CAT, GSH-Px, SOD activity	[254]
	G and/or L (oregano EO, 250 mg/kg)	milk: ↓ fat ↑ lymphocytes	=BW gain =immune response	-	[293]
	G and L (oregano EO, 15 mg/kg)	↓ TBARS at early lactation, ↑ feed intake ↑ *Lactobacillus*, ↓ *E. coli*, *Enterococcus*	↑ BW gain	-	[52]
	late G and L (garcinol, 200 or 600 mg/kg)	↑ birth BW, ↓ MDA ↑ GSH-Px, SOD, and CAT activity ↓ milk protein, ↑ IgA and IgG in colostrum and milk	↑ BW gain, ↓ mortality ↑ serum IgA and IgG	-	[294]
	late G and L (α-lipoic acid, 800 ppm)	↑ GSH-Px activity, ↓ MDA, ↑ birth BW	↑ BW gain	-	[295]
	G and L (seaweed extract, 10 g/d)	↑ IgG in colostrum; ↑ milk protein	↑ serum IgA and IgG	-	[296]
	L (silymarin, 8 g/day)	=antioxidant potential =milk composition	=BW gain	-	[314]
	late G and L (hemp seed, 2%)	↓ TBARS ↑ TAC, NO, SOD, CAT, GSH-Px	↑ SOD, CAT, GSH-Px	-	[297]
Spray-dried plasma	L (SDP, 0.25% or 0.5%)	↑ FI in ≤2 parity sows ↓ FI in >2 parity sows ↓ WEI 1st parity sows ↑ subsequent parturition rate	↑ weaning BW ↑ full value pigs at weaning in >2 parity sows	-	[308,309]
	last week of G and L (SDP, 0.5%)	↑ FI in >3 parity sows ↓ FI in ≤3 parity sows	↓ mortality in >3 parity sows	-	[311]
	late G and L (SDP, 0.5%)	=FI and =WEI	↑ weaned pigs ↓ mortality in >3 parity sows	-	[310]
	G and L (SDP, 0.5%)	↑ parturition rate ↑ piglets born alive/litter, ↑ weaned piglets/litter	-	-	[313]
	G (SDP, 0.5%)	=FI	↑ weaning BW ↑ full value pigs at weaning	-	[312]

BW = body weight; CAT = catalase; G = gestation; GSH-Px = glutathione peroxidase; EO = essential oil; FI = feed intake; HDL = high-density lipoprotein; IDO = indoleamine 2, 3-dioxygenase; Ig = immunoglobulin; IL = interleukin; L = lactation; LDL = low-density lipoprotein; MDA = Malondialdehyde; mRNA = messenger RNA; MUC2 = mucin 2; n-3 PUFA = omega-3 polyunsaturated fatty acids; NO = nitric oxide; PPARGC-α = peroxisome proliferative activated receptor gamma coactivator 1 alpha; Se = selenium; SOD = superoxide dismutase; TAC = total antioxidant capacity; TBARS = thiobarbituric acid-reactive substances; TGF-β1 = transforming growth factor beta 1; TNF-α = tumour necrosis factor alpha; WEI = weaning-to-estrus interval; (↑) = increase; (↓)= decrease; (≈) = no change.

**Table 7 animals-11-00302-t007:** Effects of probiotic supplementation of sows on piglet performance and gut health.

Probiotic	Dose	Supplementation	Performance	Gut Health	Reference
*Bacillus cereus*	0.5–1 × 10^6^ spores/g feed	Sows and piglets	↑ BW, ↑ ADG, and ↑ FCR	↓ diarrhea index	[329]
*Bacillus cereus* CIP5832 (Paciflor)	8.5 × 10^5^ cfu/g feed	Sows and piglets	↑ BW, ↑ ADG, and ↑ FCR	-	[330]
*Bacillus cereus var toyoi*	G 2.6 × 10^5^, L 4 × 10^5^,P 1.3 × 10^6^ cfu/g feed	Sows and piglets	↑ ADG, ↑ FCR	↓ diarrhea index	[326]
*Bacillus cereus var toyoi* (Toyocerin)	5 × 10^8^ spores/g feed	Sows only	↑ BW, ↓ Mortality	↓ diarrhea index	[331]
*Bacillus licheniformis* DSM5749 *+ Bacillus subtilis* DSM5750 (Bioplus 2B)	1.3 × 10^6^ spores/g feed	Sows only	↑ BW, ↓ Mortality	↓ diarrhea index	[324]
*Bacillus subtilis*	3.75 × 10^5^ cfu/g feed	Sows only	↑ BW, ↑ ADG	↑ *Lactobacillus* (ileum and colon) ↓ *E. coli* (colon), and *Clostridium perfringens* (ileum)	[327]
*Bacillus subtilis* C-3102	3 × 10^5^ cfu/g feed	Sows and piglets	↑ BW, ↑ ADG	↓ *E. coli* and *Clostridium spp.* in feces	[323]
	3 × 10^5^ cfu/g feed	Sows and piglets	-	↑ *SCFA* in distal SI and colon, ↓ villus atrophy and crypt deepening in SI	[332]
	G 5 × 10^5^, L 1 × 10^6^,P 5 × 10^5^ cfu/g feed	Sows and piglets	↓ ADG, ↓ ADFI	↑ *Bacillus spp.* counts in feces	[317]
*Bacillus subtilis* + *Lactobacillus acidophilus*	1.2 × 10^7^ + 1.15 × 10^6^ cfu/g feed	Sows only	↑ BW	-	[321]
*Bacillus mesentericus* TO-A + *Clostridium butyricum* TO-A + *Enterococcus faecalis* T-110	1 × 10^8^ + 1 × 10^8^ + 1 × 10^9^ cfu/g feed	Sows and piglets	↑ BW, ↑ FCR	↑ *Bifidobacterium* counts (ileum), ↓ diarrhea index, ↑ villus height, and ↑ villus: crypt ratio	[322]
*Enterococcus faecium NCIMB 10415*	G 1.6 × 10^6^, L 1.2 × 10^6^,P 0.17 × 10^6^ cfu/g feed	Sows and piglets	-	↓ diarrhea index	[333]
*Enterococcus faecium* DSM 7134 (Bonvital)	5 × 10^8^ cfu/g feed	Sows only	↑ BW, ↓ Pig loss	-	[334]
*Enterococcus faecium* DSM 7134	2.7–5.4 × 10^8^ cfu/kg feed	Sows only	↑ BW, ↑ ADG, and ↑ FCR	↑ *Lactobacillus* and *Enterococci* in feces, and ↓ *E. coli,* ↓ diarrhea index	[335]
*Lactobacillus helveticus BGRA43 + Lactobacillus fermentum BGHI14 + Streptococcus thermophilus BGVLJ1-44*	200 mL of mixed probiotic culture (10^8^ cfu/mL) in feed	Sows only	-	↑ Microbiota diversity, ↓ *Enterobacteriaceae* in feces	[315]
*Lactobacillus johnsonii XS4*	6.0 × 10^9^ cfu/kg feed	Sows only	↑ BW	-	[320]
*Pediococcus acidilactici* ZPA017	2.4 × 10^9^ cfu/kg feed	Sows only	↑ BW, ↓ mortality	↓ diarrhea index	[336]
*Saccharomyces cerevisiae*	1.5 × 10^10^ live cells/g feed	Sows and piglets	↑ ADG, ↑ FCR	-	[337]
*Saccharomyces cerevisiae CNCM I-4407 (Actisaf Sc47^®^)*	600 g of Actisaf Sc47^®^ per ton of feed	Sows only	↑ BW, ↓ mortality	↓ diarrhea index	[338]

ADG = average daily gain; BW = body weight; cfu = colony forming units; FCR = feed conversion ratio; G = gestation; L = lactation; P = nursing piglets; SCFA = short-chain fatty acid; SI = small intestine; (↑) = increase; (↓)= decrease; (≈) = no change.

**Table 8 animals-11-00302-t008:** Important nutrients for lactating sows and early-weaned piglets.

Nutrient	Age/duration	Dose	Results	Reference
Nucleotide	Sows day 90 gestation/40 day	0.5 to 1% of nucleotides (96.8% pure)	↑ ADFI sows, ↑ total pig born and alive, ↑ BW and ADG pigs ↑ Lactobacillus, ↓ E. coli fecal counts in sows ↓ cortisol, epinephrine, norepinephrine	[356]
	Piglets day 7 age/21 day	740.9 g nucleotides/100 kg milk replacer powder	↓ FCR, ↑ villus height, ↑ lactase, maltase, ↑ IgA, IL-1β, ↑ Claudin-1, ZO-1	[3]
	day 20 age/20 day	1.34 g nucleotides/8 mL water	↑ ADFI, ↑ IgA	[343]
Tryptophan	Sows lactation/21 day	0.17% Trp	=number and weight of piglets	[357]
	Piglets day 25 age/10 day	5 g/kg of feed	↑ Trp in plasma, hypothalamic serotonin turnover, ↑ cortisol in saliva, ↑ VH:CD, ↓ physical activity, No ≠ in gain or feed intake	[358]
	day 21 age/23 day	1 g/kg of feed	↑ ADG, FI in susceptible pigs to ETEC	[346]
	day 28 age/21 day	0.19 and 0.26% Trp	↑ ADG, FI, FCR, ↑ ghrelin: plasma and expression in gastric fundus	[347]
Leucine	day 5 of age/1–2 h	200 and 400 µmol/kg/h	↑ protein synthesis in muscle	[351]
	day 7 of age/1 h	400 µmol/kg/h for 1 h	↑ protein synthesis in muscle	[350]
	day 2 of age/21 day	800 µmol/kg/h for 1 h every 4 h	↑ BW, muscle weight, lean gain, ↑ protein synthesis, ↓ 48% fat gain	[349]
	day 21 of age/14 day	0.55% L-Leu	↑ phosphorylated levels of S6k1 and 4E-BP1 ↑ ADG, ↑ protein synthesis	[352]
	day 7 of age/14 day	1.4 g L-Leu/kg BW twice a day	↑ villus height, VH:CD ratio; ↑ BW gain ↑ plasma leucine, glutamine, and asparagine	[359]
Glutamic acid and glutamine	Sows day 90 gestation/45 day	1% Gln	↑ fetal growth, ↓ num. of IUGR piglets, ↓ variation in birth weight ↓ pre-weaning mortality of live-born piglets and IUGR; ↑ pigs and IUGR pigs growth and survival ↑ Gln in milk, plasma and skeletal muscle of sows In IUGR pigs: ↓ NH_3_ in plasma and whole-body AA oxidation	[360]
	day 85 gestation/30	1% Gln	↑ fetal growth, ↓ within-litter variation, ↓ intestinal miR-29a levels, ↑ ECM and TJ of intestine, ↓ IUGR-induced impairment↑ intestinal weight and morphology (piglets) ↑ ALP plasma levels in sows and neonatal piglets	[361]
	lactation/21 day	1–2% monosodium Glu	↑ milk production, ↑ free and peptide-bound AAs in milk ↑ growth and survival of suckling piglets ↑ efficiency of feed utilization for lactation	[362]
	day 107 gestation/28 day	1,5% Aminogut (Gln + Glu)	↑ milk Gln content, ↑ milk and colostrum fat ↑ milk somatic cell count (↑ immune function)	[363]
	day 74 gestation/61 day	2,5% Aminogut (Gln + Glu)	↑ milk Gln content ↓ fall in intramuscular Gln content during lactation	[364]
	Piglets day 7/14 day	1.52 g Gln/kg BW/day	↓ weaning stress-induced endoplasmic reticulum dysfunction and cell death in the small intestine ↓ post-weaning cytokine concentrations in the intestine	[365]
	day 7/7 day (Pigs challenged with LPS)	1 g Gln/kg BW/day	↑ piglet growth, ↑ Gln in small-intestinal lumen and plasma, ↓ expression of Toll-like receptor 4, ↓ fever, ↓ intestinal injury	[366]

ADFI = average daily feed intake; ADG = average daily gain; ALP = alkaline phosphatase; BW = body weight; 4E-BP1 = Eukaryotic translation initiation factor 4E-binding protein 1; ECM = extracellular matrix; ETEC = enterotoxigenic *Escherichia coli*; FCR = feed conversion ratio; G:F = gain to feed ratio; Gln = glutamine; Glu = glutamic acid Ig = immunoglobulin; IL = interleukin; IUGR = intra-uterine growth restriction; Leu = leucine; miR-29a = microRNA 29a; LPS = lipopolysaccharides; S6k1 = ribosomal protein S6 kinase beta-1; TJ: tight junctions; Trp = Tryptophan; VH:CD = villus height and creep depth ratio; ZO-1: zona occludens-1.

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
