# Peer review of "Management and Feeding Strategies in Early Life to Increase Piglet Performance and Welfare around Weaning: A Review"

_animals, 2021, doi:10.3390/ani11020302_

Round 1
Reviewer 1 Report
This is a practical review article on improving piglet performance through management and dietary interventions. The authors summarized many strategies that may improve the post-weaning growth, and reduce the morbidity and mortality of piglets, which are very important in pig production. Although the topic of the manuscript is of interest to the readership of the journal, the present review requires some changes. 1. Line 54, It is difficult to understand “or weight 3.3 kg less than 5.5 kg pigs” in this paragraph. 2. Line 88, “Birth BWHigh producting”. The paragraph format is incorrect. 3. Line 92, The abbreviation of “IUGR” was first appeared in line 90, so “Intra-uterine growth restriction” in line 92 should be abbreviated. 4. Line 265, ……that group housing has not negative effect…… ……that group housing has no negative effect…… 5. Line 274, …...Moreover sows that…… …...Moreover, sows that…… 6. Line 304, …staff… …staffs… 7. Line 390 and line 436, …piglet… …piglets… 8. Line 390 ……This is even worst in…… ……This is even worse in…… 9. Line 562 and line 570……Total…… ……total…… 10. Line 563-564, ……(HPr >12-14< Total born piglets)……make sure if this expression is correct. 11. Line 673……see section 4.4. In fact, there was no section 4.4 in the manuscript. 12. Line 693, …Folate, Betaine or Vitamin…… The first letter should be lowercase. As well as line 697. 13. Line 726, …Chain… …chain… 14. Line 728, …Valine… …valine… 15. Line 747, …β- oxidation… …β-oxidation… 16. Line 846, what is the abbreviation of “PA”. 17. Line 853, ……can modulate pigs’ immune response is Spray dried plasma …… ……can modulate immune response of pigs is spray dried plasma…… 18. Line 861, “Gut-Associated Lymphoid Tissue”. The first letter should be lowercase. As well as line 863 and 864. 19. In the manuscript, for unit of energy, it is better to use MJ rather than Mcal. 20. Line 901, …pig… …pigs…. 21. Line 933, ……such us is the early weaning…… ……such as the early weaning…… 22. The references style are not uniform, such as line 1048 and line 1729, which need to be checked. 23. Some pathogenic bacteria and virus infection can cause piglet diarrhea, which can also affect waning BW. It would be meaningful to summarize some strategies and dietary interventions to alleviate diarrhea and improve performance. 24. Environmental temperature is also very important for neonatal and weaning piglets. I suggested that relevant topics could be discussed appropriately.Author Response
Please see the attachment

Reviewer 2 Report
This manuscript summarized many interesting management and feeding strategies in early life to increase pig performance and welfare after weaning. This manuscript is well written and interesting. The figures accompanying the manuscript are aesthetically pleasing. The only suggestion is the author might need to check if there is any new references in this filed, especially in microbe part.
Reviewer 3 Report
This review is about the management and feeding strategies that may be applied in the early life of piglets to increase their performance and welfare around the weaning practice. The authors comprehensively summarized (reviewed) the up-to-date knowledge from the literature, although in many aspects authors still cannot not clearly tell the field nutritionists or producers what to do in practice in terms of strategy selection, especially at a point when different strategies need to be applied at the same time. That being said, this reviewer believe that the quality of this manuscript can be greatly improved upon a thorough revision with reference to the following concerns.
Major Concerns:
The title may be better as “Management and Feeding Strategies in Early Life to increase Piglet Performance and Welfare around Weaning: a Review.”
It would be better if the authors can move “Section 3.2. Feeding management” into “Section 4. Dietary interventions”.
The heading, 2.3. Microbiota, would be better to be “2.3. Gut microbiota”.
The heading, 4.1.5. Modulating microbiota, would be much better (more accurate) to be “4.1.5. Modulating microbiota with probiotics”.
Figure 1 is not clear and not easy to be understood. More details or explanation should be given.
L. 176-189: Sows’ health status and milk production should be discussed before piglet vitality and activity (see L. 169-175).
L. 283-287: “Regarding feeding management …. “ should be discussed in Section 3.2.1. Feeding sows.
L. 937-939: This sentence is not about Piglets, but about sows. It should be moved into 4.1. Sow.
L. 955-958: This sentence is not about Piglets, but about Sows. It should be moved into 4.1. Sow.
The English expression in the manuscript has a big problem. Many sentences are not grammatically correct, and many words were not appropriately used. Some examples can be seen in the following “Minor Concerns”. Please ask a native English speaker (with knowledge in swine science) to proofread the revised manuscript before resubmission.
Minor Concerns:
L 99: It should be the first 5 days. Please correct this expression throughout the manuscript.
L 117-119: Reference/citations are needed for these two sentences.
L 124-125: This sentence is not grammatically correct.
L 127-128: The sentence in the middle is not grammatically correct.
L 217: A citation is needed for this sentence.
L 238: What is “p.e.”?
L 276: hat are “The previous problems”? not clear.
L 277: We understand floor and bedding, but what is design?
L 324: and the young
L 344-345: A duration is a duration (i.e., a period of time). How could it or does it decrease sow colostrum yield?
L 486: Table 4 is not about housing!
L 588-589: This title of Table 2 should be revised.
L 592: “during lactation” is redundant.
L 665: In some places, the “first third or second third of gestation” was used, but in other places, the “first trimester or last trimester of gestation” was used. Please be consistent throughout the manuscript to avoid confusion.
L 665: What do you mean by saying “according to …”
L 673: There is no section 4.4.
L 745: you mean “expected”?
L 771: It should be “ Sows, like mammals, have ….”
L 773-774: This sentence sound grammatically incorrect.
L 851: enhanced composition? How?
L 866-867: “It has been reported an increase in litter and individual pig weight at weaning” sounds awkward.
L 888 some what translated?
L 890-894: This sentence is way too long. Nobody could understand it.
L 993: such us is?
L 947-948: “when Leu is supplemented in low-protein diets, increases protein synthesis in muscle and organs, and daily gain weight”. What subject increases ….?
L 964: for lactating sows and …
L 971: evidenced?
References (from L 992): All the article titles should use “Sentence case”.
By the way, do we think too many articles/books (a total of 375 publications across almost 50 years from 1974 to 2020) were cited in the paper? Maybe we can reduce the number of articles. Some old articles may be too old to be used.
Round 2
Reviewer 3 Report
Major Concerns:
The title may be better as “Management and Feeding Strategies in Early Life to Improve Piglet Performance and Welfare around Weaning: a Review.”
Overall, it is Management Strategies that should be applied by humans to improve pigs’ performance and welfare. Feeding/dietary/nutritional strategies are only one category of Management Strategies. In this paper, the feeding/nutritional strategies are separated from the Management Strategies, and the two categories of strategies are discussed, according to the title, the summary, the abstract, the introduction, and the conclusioms. Therefore, Section “3.2. Feeding management’ should be moved into Section “4. Nutritional interventions”. Feeding management, dietary management, and nutritional interventions are all the same thing; all belong to the category of feeding, dietary, or nutritional strategies (whatever you call it).
Minor Concerns:
L 17: for enhancing
L 83-85: This sentence is not correct.
L 313: “design” should be “barn design”.
L 1041: “such us” or “such as”?
Hope we can have a clean copy to quickly read through.